# Regioselective and Stereoselective Synthesis of Parthenolide Analogs by Acyl Nitroso-Ene Reaction and Their Biological Evaluation against *Mycobacterium tuberculosis*

**DOI:** 10.3390/ijms242417395

**Published:** 2023-12-12

**Authors:** Bruna Gioia, Francesca Ruggieri, Alexandre Biela, Valérie Landry, Pascal Roussel, Catherine Piveteau, Florence Leroux, Ruben C. Hartkoorn, Nicolas Willand

**Affiliations:** 1Univ. Lille, Inserm, Institut Pasteur de Lille, U1177—Drugs and Molecules for Living Systems, F-59000 Lille, Francefrancesca.ruggieri@univ-lille.fr (F.R.); alexandre.biela@pasteur-lille.fr (A.B.); valerie.landry@pasteur-lille.fr (V.L.); catherine.piveteau@univ-lille.fr (C.P.); florence.leroux@pasteur-lille.fr (F.L.); 2Univ. Lille, CNRS, ENSCL, Centrale Lille, Univ. Artois, UMR 8181—UCCS—Unité de Catalyse et de Chimie du Solide, F-59000 Lille, France; pascal.roussel@univ-lille.fr; 3Univ. Lille, CNRS, Inserm, CHU Lille, Institut Pasteur de Lille, U1019—UMR 8204—CIIL—Center for Infection and Immunity of Lille, F-59000 Lille, France; ruben.hartkoorn@inserm.fr

**Keywords:** natural products, semisynthesis, *Mycobacterium tuberculosis*

## Abstract

Historically, natural products have played a major role in the development of antibiotics. Their complex chemical structures and high polarity give them advantages in the drug discovery process. In the broad range of natural products, sesquiterpene lactones are interesting compounds because of their diverse biological activities, their high-polarity, and sp^3^-carbon-rich chemical structures. Parthenolide (PTL) is a natural compound isolated from *Tanacetum parthenium*, of the family of germacranolide-type sesquiterpene lactones. In recent years, parthenolide has been studied for its anti-inflammatory, antimigraine, and anticancer properties. Recently, PTL has shown antibacterial activities, especially against Gram-positive bacteria. However, few studies are available on the potential antitubercular activities of parthenolide and its analogs. It has been demonstrated that parthenolide’s biological effects are linked to the reactivity of *α*-exo-methylene-*γ*-butyrolactone, which reacts with cysteine in targeted proteins via a Michael addition. In this work, we describe the ene reaction of acylnitroso intermediates with parthenolide leading to the regioselective and stereoselective synthesis of new derivatives and their biological evaluation. The addition of hydroxycarbamates and hydroxyureas led to original analogs with higher polarity and solubility than parthenolide. Through this synthetic route, the Michael acceptor motif was preserved and is thus believed to be involved in the selective activity against *Mycobacterium tuberculosis*.

## 1. Introduction

Natural products (NPs) and their derivatives have always been an important source of new medicines. Natural products often have complex chemical structures and can exhibit a wide range of biological activities, making them valuable starting points in the drug discovery process. Compared with conventional synthetic molecules, NPs are sometimes also more polar, which is an advantage in the discovery of new targets [1]. In the vast range of NPs, sesquiterpene lactones are compounds of interest for their structural complexity and diversity. Parthenolide (PTL) is a germacranolide-type sesquiterpene lactone isolated from *Tanacetum parthenium* [2]. For decades, PTL has been known for its anti-inflammatory and antimigraine effects [3]. The wide range of biological activities shown by PTL is mainly attributed to the *α*-exo-methylene-*γ*-butyrolactone moiety present in its structure [4,5]. In fact, *α*-exo-methylene is able to react with nucleophilic cysteines from biomolecules through a Michael addition (Figure 1) [6,7,8] to form a covalent bond.

Through this mechanism, PTL inhibits irreversibly the NF-*κ*B pathway, which is involved in anti-inflammatory activity, in the regulation of cell proliferation, and in anti-apoptotic mechanisms [9]. Parthenolide reacts with thiols from glutathione, leading to the depletion of GSH and consequent activation of the ROS cascade [10,11]. Precisely for all these targets, parthenolide is well-known as an anticancer agent. Many studies have been carried out through the chemical modification of PTL, mainly focusing on α-*exo*-methylene, including (i) a Michael addition with primary and secondary amines and thiol groups with increasing solubility [12,13,14,15] and (ii) Heck coupling with aromatic iodo-compounds [16]. All these studies have shown important advances in the anticancer properties of parthenolide [11,17]. More recently, interest in PTL as an antibacterial agent has grown. Hakkinen et al. studied the activity against Gram-positive and Gram-negative bacteria, and PTL showed higher activity against *Staphylococcus aureus* (*S. aureus*) than against Gram-negative bacteria, such as *Pseudomonas aeruginosa* and *Escherichia coli* [18,19]. Few studies have investigated the activity against *Mycobacterium tuberculosis* (*M. tuberculosis*) [20,21,22]. PTL displayed a minimum inhibitory concentration (MIC) of 16 µg/mL against *M. tuberculosis* strains, but no further investigation has been carried out on the structure–activity relationship or target identification [22]. Hence, we planned the synthesis of original parthenolide-based derivatives using an acylnitroso-ene reaction, which were evaluated as antibacterial agents and, more precisely, as anti-tuberculosis drugs. With this work, we optimize a regioselective and stereoselective synthetic route for original analogs. Through an acyl nitroso-ene modification, the Michael acceptor motif is preserved. Moreover, owing to the introduction of the hydroxylamine moiety on the main core, the new semisynthetic analogs displayed increased solubility compared to that of the parent compound.

## 2. Results and Discussion

### 2.1. Chemistry

C-7-hydroxycarbamate/urea–parthenolide analogs were synthesized from commercially available parthenolide via a reaction with hydroxylamine-functionalized building blocks (Figure 1). First, the optimization was performed, starting from the conditions previously described by Frazier et al. [23]. The critical step in the acylnitroso-ene reaction is the oxidation of hydroxylamine-containing compounds to nitroso derivatives prior to the reaction with parthenolide.

Therefore, an optimization step was necessary to identify the best conditions and the best oxidative system. Different reaction times, equivalents, and oxidant agents were investigated in a model reaction with 1-hydroxy-3-phenyl-urea (**2a**). All the studied conditions are available in the Appendix A. After the optimization, oxygen was chosen as oxidizing agent, and the reaction was carried out in the presence of a catalyst system of CuCl (0.05 eq.), pyridine (0.125 eq.), and THF as the solvent (Figure 2).

To expand the scope of this reaction, two series of compounds were synthesized based on hydroxylamine-containing functions: hydroxyurea and hydroxycarbamate, either purchased or synthesized. Urea (**2a**–**h**) and carbamate (**3a**–**k**) precursors were obtained from isocyanates and chloroformates, respectively (Figure 3A). Unfortunately, the introduction of more hydrophilic moieties was challenging owing to the high polarity of the corresponding ureas. In fact, after the work-up, urea precursors were isolated as a mixture of the expected compound, salts, and HCl residues, which prevented the formation of nitroso-ene compounds. On the other hand, ureas bearing aromatic substituents were easily accessed. Hydroxycarbamates with polar and less-polar groups were easily obtained in good to high yields. Parthenolide reacted with a variety of substrates, from some polar alkyl groups to more lipophilic aromatic substituents (Figure 3B) under the optimized conditions in Figure 2. As a result, nineteen original compounds were obtained in yields ranging from 25 to 91%.

Moreover, it must be noted that the reaction between the parthenolide and hydroxyurea led to the formation of two compounds (Figure 4). In fact, it was observed that after 48 h, a mixture of the parthenolide, hydroxyurea derivative (**9**), and side compound corresponding to the oxime derivative (**23**) was obtained. Compound **23** was then isolated, and a biological assessment was carried out. Surprisingly, the removal of the O_2_ supply resulted in the formation of hydroxyurea (**9**) in a very good yield, and just traces of oxime were observed.

Conditions under air were tested for this reaction, and it was observed that O_2_ enhanced the acylnitroso-ene reaction, and no formation of oxidized compounds was observed (entry E, Appendix A). To the best of our knowledge, this is the first report of a nitroso-ene reaction with parthenolide under mild conditions and with high stereoselectivity and regioselectivity to give twenty semisynthetic analogs in moderate to good yields.

#### 2.1.1. Proposed Reaction Mechanism

Numerous hypotheses have been postulated for the mechanism of the nitroso-ene reaction, but it has not been totally elucidated yet [24,25,26,27,28]. The ene reaction of nitrosocarbonyl derivatives follows Markovnikov’s rule with the addition of the electron-deficient enophile (**24**) to the less-substituted carbon of the alkene (**1**). According to previous studies, we hereby suggest that the reaction takes place through an intermediary aziridinium state (**25**, Figure 5), which can explain the regioselectivity and stereoselectivity. In an oxidant environment, O_2_ leads to oxidation from Cu (I) to Cu (II), then Cu (II) oxidizes the hydroxylamine derivative (**24**, hydroxyurea or hydroxycarbamate) to nitrosocarbonyl species. The combination of CuCl, pyridine, and THF promotes the generation of the transient state of nitroso species in situ [29]. Afterward, a concerted mechanism takes place: the electrophilic nitrogen of **24** attacks the ene species, parthenolide, and, vice versa, the ene attacks the enophile (**24**) leading to the aziridium intermediate (**25**). The intermediate (**25**), which is unstable, evolves to the final compound by H-abstraction from the more-substituted and less-crowded end, leading to high regioselectivity. The less-hindered *Re*-face attack from the nitroso species is favored, thus conferring high stereoselectivity.

#### 2.1.2. X-ray Crystallographic and NOESY Data

In support of our hypothesis, the configuration of the new chiral carbon (C-7) was determined by a NOESY experiment with compound **14**. MOE modelling enabled the determination of four possible structures: two for each configuration *R* or *S* (images are available in Appendix A). For the *R* configuration, both models showed a possible spatial interaction between proton 3 and proton 7 (depicted in green, Figure 2A). On the contrary, for the *S* configuration, the two models showed that proton 7 is in spatial proximity to methyl group 15 (depicted in red, Figure 2B). NOESY spectra determined correlations between H3/H7 and H2/H15, suggesting the *R* configuration for carbon 7. In addition, obtaining the crystallographic structure of **14** enabled us to confirm the exact configuration (Figure 2C).

### 2.2. Biological Evaluation

All the synthesized compounds together with parthenolide were tested against the virulent strain *M. tuberculosis* H37Rv and *S. aureus* (Table 1 and Table 2). A resazurin microtiter assay (REMA) and a GFP fluorescent assay were performed for *M. tuberculosis*, and a REMA assay was performed for *S. aureus*. The compounds were dissolved in DMSO and tested at eight different concentrations (from 300 µM to 2.3 µM). After a period of incubation (7 days for *M. tuberculosis* and 5 h for *S. aureus*), resazurin was added, and the fluorescence was measured. Rifampicin and DMSO were used as positive and negative controls, respectively. The results were expressed as MIC_90_ (REMA assay) or MIC_98_ (GFP assay) values, meaning that the drug concentration inhibits 90% or 98% of the bacterial growth. All the nitroso-ene derivatives showed antibacterial activity against *M. tuberculosis*. Instead, no activity (MIC_90_ ca. 300 µM) was observed against *S. aureus*. It was observed that bulky groups enhance the activity against *M. tuberculosis*. In fact, compounds **9** and **23** with a hydroxyurea and an oxime moiety, respectively, were found to be inactive (MIC_90_ ca. 150 µM). In accordance with this observation, aromatic substituents were well-tolerated for both series. However, the urea derivatives showed greater inhibitory activity against *M. tuberculosis* strains. The introduction of small substituents, such as a methoxy group (compound **7,**
Table 1) and a fluorine atom (compound **8,**
Table 1), to the para position of the benzene ring enhances the inhibitory activity with a MIC_90_ value of 19 µM (REMA assay). Compared with the parent compound, the most active compounds are in the same range of activity (MIC_90_ is between 9.4 µM and 19 µM). However, advantageously, all the nitroso-ene derivatives are more polar and more water soluble than PTL. The urea derivatives (compounds **8** and **9**) showed from 1.3- to 1.5-fold higher solubility in PBS compared to parthenolide, respectively.

As described in the introduction (Section 1), parthenolide reacts with biological nucleophiles through a Michael addition, and it results in potent anticancer agents. Given their unspecific electrophilicity, all the synthesized compounds were also tested for their cytotoxicity activities in Balb3T3 cell lines. The aromatic derivatives showed increased cytotoxicities compared with that of the parent compound. However, the specific activity against *M. tuberculosis* does not seem to be related to cytotoxicity, given the absence of activity against *S. aureus*. At this stage of the investigation, we can only suggest that the Michael acceptor is involved in the activity. In line with our hypothesis, no activity against either *M. tuberculosis* or *S. aureus* was found for Dihydro-PTL (**26,**
Table 2). Extensive studies are needed to discover a possible target and/or perhaps a mechanism of action.

## 3. Materials and Methods

### 3.1. Chemistry

The reagents and solvents for synthesis, analysis, and purification were purchased from commercial suppliers (Fisher Scientific (France), Sigma Aldrich (France), Key Organics (UK), and Fluorochem (UK and Ireland)) and used without further purification. The progress of all the reactions was routinely monitored by thin-layer chromatography (TLC) and/or by high-performance liquid chromatography–mass spectrometry (HPLC–MS). TLC was performed using Merck^®^ commercial aluminum sheets coated with silica gel 60 F254. Visualization was achieved by fluorescence quenching under UV light at 254 nm and 215 nm or by staining with potassium permanganate. Purifications were performed by flash chromatography and reverse chromatography. Flash chromatography purifications were conducted in columns prepacked with Reveleris^®^ flash cartridges, Buchi, France (40 μm, Büchi^®^ FlashPure, Villebon sur Yvette, France or 15–40 μM, Macherey-Nagel^®^ Chromabond^®^, Hoerdt, France) under pressure with an Interchim Puriflash^®^ 430 instrument (Montluçon, France). The products were detected by UV absorption at 254 nm and by ELSD. Reverse chromatographies were conducted using Combiflash^®^ C18 Rf200 in 4g, 12g, and C18 columns (Serlabo Technologies, EntraiguessurlaSorgues, France). The products were detected by UV absorption at 215 nm and 254 nm. UPLC–MS analysis was performed on an LC–MS Waters ACQUITY UPLC I-Class system (Guyancourt, France) equipped with a UPLC I BIN SOL MGR solvent manager, a UPLC I SMP MGR-FTN sample manager, an ACQUITY UPLC I-Class eλ PDA photodiode array detector (210–400 nm), and an ACQUITY QDa (Performance) mass detector (full scan ESI+/− in the range 30–1250). A Waters Acquity BEH C18 column (1.7 μm particle size; dimensions: 50 mm × 2.1 mm) was used for UPLC analysis (Guyancourt, France). The injection volume was 0.5 μL. For a 5 min analysis, the elution was carried out with a gradient starting at 98% H_2_O and 5 mM ammonium formate (pH 3.8) and reaching 98% CH_3_CN and 5 mM ammonium formate (pH 3.8; 5% aqueous) over 3.5 min at a flow rate of 600 µL/min. For a 30 min analysis, the elution was carried out with a gradient starting at 98% H_2_O and 5 mM ammonium formate (pH 3.8) and reaching 98% CH_3_CN and 5 mM ammonium formate (pH 3.8; 5% aqueous) over 25 min at a flow rate of 600 µL/min. Purity (%) was determined by reverse-phase UPLC, using UV detection (at 215 and 254 nm). HRMS analyses were performed on an LCT Premier XE Micromass (Guyancourt, France), using a Waters XBridge BEH C18 column with a 3.5 μm particle size and dimensions of 50 mm × 4.6 mm. A gradient starting from 98% H_2_O and 5 mM ammonium formate (pH 3.8) and reaching 100% CH_3_CN and 5 mM ammonium formate (pH 3.8; 5% aqueous) within 3 min at a flow rate of 2 mL/min was used. NMR spectra were recorded on a Bruker^®^ Avance-300 spectrometer (Wissembourg, France). The results were calibrated to signals from the solvent as an internal reference (e.g., 7.26 (residual CDCl_3_) and 77.16 (CDCl_3_) ppm and 2.50 (residual DMSO d^6^) and 39.52 (DMSO d^6^) ppm for ^1^H and ^13^C NMR spectra, respectively.) Chemical shifts (δ) are in parts per million (ppm) downfield from tetramethylsilane (TMS). The assignments were made using one-dimensional (1D) ^1^H and ^13^C spectra and two-dimensional (2D) HSQC-DEPT, COSY, and HMBC spectra. NMR coupling constants (J) are reported in hertz (Hz), and splitting patterns are indicated as follows: s (singlet); brs (broad singlet); d (doublet); dd (doublet of doublet); ddd (double of doublet of doublet); dt (doublet of triplet); t (triplet); td (triplet of doublet); q (quartet); m (multiplet).

#### 3.1.1. General Procedure for the Synthesis of Hydroxyurea Derivatives (**2a**–**h**)

In a round-bottom flask, 1.0 eq. of isocyanate was diluted in DCM. Then, 6.0 eq. of hydroxylamine hydrochloride and 6.0 eq. of sodium hydroxide in water were added at 0 °C, and the solution was stirred for 3 h at room temperature. The aqueous layer was extracted once with DCM. The dichloromethane phase was discarded. Then, the aqueous phase was acidified with HCl (1 M) and extracted three times with ethyl acetate. The organic layer was dried over MgSO_4_ and concentrated under reduced pressure to give a crude residue. Purification by flash chromatography afforded the desired compound.

1-hydroxy-3-phenyl-urea (**2a**) was obtained as a white powder, without further purification (190 mg, quant.). ^1^H NMR (300 MHz, DMSO-d^6^) δ = 8.93 (s, 1H); 8.80 (s, 1H); 8.73 (s, 1H); 7.59 (dd, J = 1.11, 8.69 Hz, 2H); 7.23 (t, J = 7.28 Hz, 2H); 6.95 (t, J = 7.28 Hz, 1H). NMR data are in agreement with previously published data [26]. 

1-hydroxy-3-(p-tolyl)urea (**2b**) was obtained as a white powder after purification by flash chromatography in column 15g, with a gradient from 100% dichloromethane to 70/30 dichloromethane/ethyl acetate (257 mg, 41%). ^1^H NMR (300 MHz, DMSO-d^6^) δ = 8.89 (s, 1H); 8.73 (s, 1H); 8.63 (s, 1H); 7.47 (m, 2H); 7.04 (d, J = 8.38 Hz, 2H); 2.22 (s, 3H). ^13^C NMR (75 MHz, DMSO-d^6^) δ = 158.70, 136.76, 130.90, 128.84 (2xC-Ar); 119.25 (2xC-Ar); 20.36.

1-benzyl-3-hydroxy-urea (**2c**) was obtained as a white powder after purification by reversed-phase chromatography in column 15g, with a gradient from water/acetonitrile (90/10) to acetonitrile 100% (154 mg, 25%). ^1^H NMR (300 MHz, DMSO-d^6^) δ = 8.63 (bs, 1H); 8.38 (s, 1H); 7.17–7.32 (m, 6H); 4.23 (d, J = 6.34 Hz, 2H). NMR data are in agreement with previously published data [26]. 

1-hydroxy-3-(4-methoxyphenyl)urea (**2d**) was obtained as a white powder, without further purification (116 mg, 41%). ^1^H NMR (300 MHz, DMSO-d^6^) δ = 8.85 (d, J = 0.85 Hz, 1H); 8.67 (s, 1H); 8.61 (s, 1H); 7.48 (m, 2H); 6.82 (m, 2H); 3.69 (s, 3H). ^13^C NMR (75 MHz, DMSO-d^6^) δ = 158.85; 154.59; 132.37; 120.87; 113.59; 55.11.

1-(4-fluorophenyl)-3-hydroxy-urea (**2e**) was obtained as a slightly yellow powder, without further purification (366 mg, 82%). ^1^H NMR (300 MHz, DMSO-d^6^) δ = 8.93 (s, 1H); 8.85 (s, 1H); 8.82 (s, 1H); 7.62 (m, 2H); 7.07 (m, 2H). 19F NMR (300 MHz, DMSO-d6) δ = −121.3. ^13^C NMR (75 MHz, DMSO-d^6^) δ = 158.70, 157.61 (d, J = 239.22 Hz); 135.78 (d, J = 2.40 Hz); 120.92 (d, J = 7.67 Hz, 2XC-Ar); 114.89 (d, J = 21.90 Hz, 2XC-Ar).

1-(1,3-benzodioxol-5-yl)-3-hydroxy-urea (**2g**) was obtained as a white powder after purification by flash chromatography in column 15g, with a gradient from 100% dichloromethane to 70/30 dichloromethane/ethyl acetate (108 mg, 90%). ^1^H NMR (300 MHz, DMSO-d^6^) δ = 8.87 (d, J = 0.73 Hz, 1H); 8.72 (s, 1H); 8.66 (s, 1H); 7.28 (d, J = 2.09 Hz, 1H); 7.02 (dd, J = 8.36, 2.09 Hz, 1H); 6.78 (d, J = 8.47 Hz, 1H); 5.94 (s, 2H). ^13^C NMR (75 MHz, DMSO-d^6^) δ = 158.70; 146.88; 142.14; 133.74; 111.95; 107.76; 101.64; 100.70.

1-(3-ethoxypropyl)-3-hydroxy-urea (**2h**) was obtained as a white powder after purification by reversed-phase chromatography in column 15g, with a gradient from water/acetonitrile (90/10) to acetonitrile 100% (194 mg, 82%). ^1^H NMR (300 MHz, DMSO-d^6^) δ = 8.54 (bs, 1H); 8.22 (s, 1H); 6.68 (s, J = 5.79 Hz, 1H); 3.40 (m, 2H, under solvent peak); 3.09 (q, J = 6.84 Hz, 2H); 1.63 (m, 2H); 1.12–1.07 (t, J = 6.90 Hz, 5H). ^13^C NMR (75 MHz, DMSO-d^6^) δ = 158.88; 67.55; 65.27; 36.52; 30.28; 15.15.

#### 3.1.2. General Procedure for the Synthesis of Hydroxycarbamate Derivatives (**3a**–**k**)

In a round-bottom flask, 1.0 eq. of chloroformate was added to a stirred solution of hydroxylamine hydrochloride (2.50 eq.) and sodium hydroxide (3 eq.) in water at room temperature. After 4 h, the mixture was acidified with HCl (1 M) and then extracted three times with ethyl acetate. The organic layer was dried over MgSO_4_ and concentrated under reduced pressure to give a crude residue. Purification by flash chromatography afforded the desired compound.

Allyl N-hydroxycarbamate (**3b**) was obtained as a clear oil after purification by flash chromatography in column 15g, with a gradient from 100% cyclohexane to 40/60 cyclohexane/ethyl acetate (93 mg, 95%). ^1^H NMR (300 MHz, DMSO-d^6^) δ = 9.63 (bs, 1H); 8.71 (s, 1H); 5.89 (m, 1H); 5.22 (m, 2H); 4.50 (dt, J = 5.34, 1.48 Hz). NMR data are in agreement with previously published data [30].

2,2,2-trichloroethyl N-hydroxycarbamate (**3d**) was obtained as a clear oil after purification by flash chromatography in column 15g, with a gradient from 100% cyclohexane to 40/60 cyclohexane/ethyl acetate (373 mg, 71%). ^1^H NMR (300 MHz, DMSO-d^6^) δ = 10.25 (bs, 1H); 9.01 (s, 1H); 4.82 (s, 2H). ^13^C NMR (75 MHz, DMSO-d^6^) δ = 155.60; 96.08; 73.20. 

4-(fluorophenyl) N-hydroxycarbamate (**3e**) was obtained as a white powder, without further purification (337 mg, 86%). ^1^H NMR (300 MHz, DMSO-d^6^) δ = 10.33 (bs, 1H); 9.08 (s, 1H); 7.11–7.24 (m, 4H). NMR data are in agreement with previously published data [31].

Phenyl N-hydroxycarbamate (**3f**) was obtained as a white powder, without further purification (331 mg, 91%). ^1^H NMR (300 MHz, DMSO-d^6^) δ = 10.29 (bs, 1H); 9.09 (s, 1H); 7.39 (t, J = 7.73 Hz, 2H); 7.21 (t, J = 7.73 Hz, 1H); 7.10 (d, J = 7.73 Hz, 2H). NMR data are in agreement with previously published data [26].

Isobutyl N-hydroxycarbamate (**3g**) was obtained as a white powder after purification by flash chromatography in column 15g, with a gradient from 100% dichloromethane to 70/30 dichloromethane/ethyl acetate (145 mg, 63%). ^1^H NMR (300 MHz, DMSO-d^6^) δ = 9.52 (bs, 1H); 8.63 (s, 1H); 3.76 (d, J = 6.72 Hz, 2H); 1.83 (sept, J = 6.59 Hz, 1H); 0.86 (d, J = 6.59 Hz, 6H). ^13^C NMR (75 MHz, CDCl_3_) δ = 158.03; 69.89; 27.65; 18.82 (2xC).

Ethyl N-hydroxycarbamate (**3h**) was obtained as a white powder, without further purification (120 mg, 72%). ^1^H NMR (300 MHz, DMSO-d^6^) δ = 9.5 (bs, 1H); 8.63 (d, J = 1.24 Hz, 1H); 4.02 (q, J = 7.07 Hz, 2H); 1.15 (t, J = 7.07 Hz, 3H). NMR data are in agreement with previously published data [32].

1,3-benzodioxol-5-yl N-hydroxycarbamate (**3i**) was obtained as a white powder after purification by flash chromatography in column 15g, with a gradient from 100% dichloromethane to 70/30 dichloromethane/ethyl acetate (75 mg, 51%). ^1^H NMR (300 MHz, DMSO-d^6^) δ = 10.23 (bs, 1H); 9.03 (s,1H); 6.88 (d, J = 8.29 Hz, 1H); 6.74 (d, 1H, J = 2.42 Hz); 6.54 (dd, 1H, J = 2.42, 8.29 Hz); 6.04 (s, 2H). ^13^C NMR (75 MHz, DMSO-d^6^) δ = 155.70; 147.47; 144.98; 144.46; 113.99; 107.79; 103.92; 101.58.

2-methoxyethyl N-hydroxycarbamate (**3j**) was obtained as an oily white solid, without further purification (230 mg, quant.). ^1^H NMR (300 MHz, DMSO-d^6^) δ = 9.61 (bs, 1H); 8.67 (d, J = 1.19 Hz, 1H); 4.09 (m, 2H); 3.48 (m, 2H); 3.24 (s, 3H). ^13^C NMR (75 MHz, DMSO-d^6^) δ = 157.71; 70.19; 63.13; 57.96.

Tetrahydropyran-4-yl N-hydroxycarbamate (**3k**) was obtained as a white powder, without further purification (100 mg, 95%). ^1^H NMR (300 MHz, DMSO-d^6^) δ = 9.56 (bs, 1H); 8.65 (s, 1H); 4.71 (sept, J = 4.29, Hz, 1H); 3.79 (m, 2H); 3.41 (m, 2H); 1.84 (m, 2H); 1.48 (m, 2H). ^13^C NMR (75 MHz, DMSO-d^6^) δ = 157.59; 69.58; 65.09 (2xC); 32.49 (2xC).

#### 3.1.3. General Procedure for the Synthesis of Acylnitroso-Ene Derivatives (**4**–**22**)

In a double-neck flask, CuCl (0.05 eq., 1 mg) and pyridine (0.125 eq., 2 µL) were added to the parthenolide (50 mg) and hydroxylamine derivative (1.1 eq.). Then, 2.2 mL of THF was added. The reaction mixture was stirred at room temperature under O_2_ (in a balloon). The reaction was monitored using TLC. After 24 h or 48 h, the reaction was stopped with a solution of EDTA (0.5 M, pH 7.0), diluted with ethyl acetate, and stirred until the color no longer persisted in the organic layer. Then, the water phase was extracted three times withe ethyl acetate. The organic phase was dried over MgSO_4_ and concentrated under reduced pressure to give a crude residue. Purification by flash chromatography afforded the desired compound.

1-hydroxy-1-[(1S,2S,4R,7R,11S)-4-methyl-8,12-dimethylene-13-oxo-3,14-dioxatricyclo[9.3.0.02,4] tetradecan-7-yl]-3-phenyl-urea (**4**) was obtained as a white powder after purification by flash chromatography in column 4g, with a gradient from 100% dichloromethane to 50/50 dichloromethane/ethyl acetate over 30 min (64 mg, 80%, purity at 254 nm: 100%). ^1^H NMR (300 MHz, DMSO-d^6^) δ = 9.42 (s, 1H, OH); 8.90 (s, 1H, NH); 7.59 (dd, J = 1.17, 8.76 Hz, 2H, 2 Ar-ortho); 7.23 (t, J = 7.47 Hz, 2H, 2 Ar-meta); 6.96 (m, 1H, Ar-para); 6.03 (d, J = 3.40 Hz, H-11); 5.64 (d, J = 3.12 Hz, 1H, H-11); 5.32 (s, 1H, H-14); 5.08 (s, 1H, H-14); 4.71 (dd, J = 2.84, 11.07 Hz, 1H, H-7); 3.97 (t, J = 9.20 Hz, 1H, H-2); 3.25 (m, 1H, H-1); 2.94 (d, J = 8.51 Hz, 1H, H-3); 2.14–2.38 (m, 5H, H-5, 6, 9, 10); 1.64–1.75 (m, 2H, H-10,6); 1.35 (s, 3H, H-15); 1.07 (t, 1H, H-5). ^13^C NMR (75 MHz, DMSO-d^6^) δ = 169.84 (Cq, C-13); 157.62 (Cq, C-16); 144.81 (Cq, C-12); 140.28 (Cq, C-8); 139.83 (Cq, C-17); 128.84 (CH, 2 C-Ar meta); 122.73 (CH, C-para Ar); 119.80 (CH, 2 C-Ar ortho); 119.61 (CH_2_, C-11); 116.22 (CH_2_, C-14); 80.18 (CH, C-2); 64.20 (CH, C-7); 63.18 (CH, C-3); 60.87 (Cq, C-4); 44.28 (CH, C-1); 36.46 (CH_2_, C-5); 29.82 (CH_2_, C-9); 26.12 (CH_2_, C-6); 25.09 (CH2, C-10); 18.15 (CH_3_, C-15). HRMS (TOF, ES+) *m*/*z* [M+H]^+^: calcd. for C_22_H_27_N_2_O_5_ 399.1920; found 399.1940. 

1-hydroxy-1-[(1S,2S,4R,7R,11S)-4-methyl-8,12-dimethylene-13-oxo-3,14-dioxatricyclo[9.3.0.02,4]tetradecan-7-yl]-3-(p-tolyl)urea (**5**) was obtained as a white powder after purification by flash chromatography in column 4g, with a gradient from 100% dichloromethane to 70/30 dichloromethane/ethyl acetate over 30 min (71 mg, 85%, purity at 254 nm: 96%). ^1^H NMR (300 MHz, DMSO-d^6^) δ = 9.36 (s, 1H, OH); 8.79 (s, 1H, NH); 7.46 (d, J = 8.32 Hz, 2H ortho); 7.03 (d, J = 8.13 Hz, 2H meta); 6.03 (d, J = 3.43 Hz, 1H, H-11′); 5.64 (d, J = 3.09 Hz, 1H, H-11); 5.32 (s, 1H, H-14′); 5.07 (s, 1H, H-14); 4.69 (dd, J = 2.83, 11.16 Hz, 1H, H-7); 3.97 (t, J = 9.27 Hz, 1H, H-2); 3.25 (m, 1H, H-1); 2.93 (d, J = 8.70 Hz, 1H, H-3); 2.13–2.37 (m, 8H, H-5′, H-6′, H10′, H-9, H-21); 1.64–1.72 (m, 2H, H-10, H-6); 1.35 (s, 3H, H-15); 1.06 (t, J = 12.11 Hz, 1H, H-5). ^13^C NMR (75 MHz, DMSO-d^6^) δ = 169.39 (Cq, C-13); 157.26 (Cq, C-16); 144.39 (Cq, C-8); 139.84 (Cq, C-12); 136.80 (Cq, C-17); 131.10 (Cq, C-20); 128.80 (CH, C-Ar meta); 119.42 (CH, C-Ar ortho); 119.17 (CH2, C-11); 115.73 (CH2, C-14); 79.74 (CH, C-2); 63.79 (CH, C-7); 62.74 (CH, C-3); 60.43 (Cq, C-4); 43.84 (CH, C-1); 36.02 (CH_2_, C-5); 29.40 (CH_2_, C-9); 25.66 (CH_2_, C-6); 24.67 (CH_2_, C-10); 20.36 (CH_3_, C-21); 17.70 (CH_3_, C-15). HRMS (TOF, ES+) *m*/*z* [M+NH_4_]^+^: calcd. for C_23_H_29_N_2_O_5_ 413.2096; found 413.2076. 

3-benzyl-1-hydroxy-1-[(1S,2S,4R,7R,11S)-4-methyl-8,12-dimethylene-13-oxo-3,14-dioxatricyclo [9.3.0.02,4]tetradecan-7-yl]urea (**6**) was obtained as a white powder after purification by flash chromatography in column 4g, with a gradient from 100% dichloromethane to 70/30 dichloromethane/ethyl acetate over 30 min (50 mg, 60%, purity at 254 nm: 100%). ^1^H NMR (300 MHz, DMSO-d^6^) δ = 9.01 (s, 1H, OH); 7.44 (t, J = 6.16 Hz, 1H, NH); 7.17–7.32 (m, 5H, H-Ar); 6.03 (d, J = 3.36 Hz, 1H, H-11); 5.65 (d, J = 3.17 Hz, 1H, H-11′); 5.27 (s, 1H, H-14); 5.04 (s, 1H, H-14′); 4.58 (dd, J = 3.17, 11.38 Hz, 1H, H-7); 4.22 (d, J = 6.34 Hz, 2H, H-17); 3.96 (t, J = 9.14 Hz, 1H, H-2); 3.24 (m, 1H, H-1); 2.90 (d, J = 8.77 Hz, 1H, H-3); 2.11–2.33 (m, 5H, H-5′, 6′, 9, 9′, 10′); 1.61–1.72 (m, 2H, H-6, 10); 1.33 (s, 3H, H-15); 1.02 (t, J = 13.07 Hz, 1H, H-5). ^13^C NMR (75 MHz, DMSO-d^6^) δ = 169.39 (Cq, C-13); 160.40 (Cq, C-16); 144.53 (Cq, C-8); 140.62 (Cq, C-18); 139.85 (Cq, C-12); 128.07 (CH, 2xC-Ar meta); 127.01 (CH, 2xC-Ar ortho); 126.48 (CH, C-Ar para); 119.16 (CH_2_, C-11); 115.48 (CH_2_, C-14); 79.75 (CH, C-2); 64.49 (CH, C-7); 62.74 (CH, C-3); 60.44 (Cq, C-4); 43.81 (CH, C-1); 42.84 (CH_2_, C-17); 36.05 (CH_2_, C-5); 29.47 (CH_2_, C-9); 25.50 (CH_2_, C-6); 24.67 (CH_2_, C-10); 17.69 (CH_3_, C-15). HRMS (TOF, ES+) *m*/*z* [M+H]^+^: calcd. for C_23_H_29_N_2_O_5_ 413.2076; found 413.2081.

1-hydroxy-3-(4-methoxyphenyl)-1-[(1S,2S,4R,7R,11S)-4-methyl-8,12-dimethylene-13-oxo-3,14-dioxatricyclo[9.3.0.02,4]tetradecan-7-yl]urea (**7**) was obtained as a white powder after purification by flash chromatography in column 4g, with a gradient from 100% dichloromethane to 70/30 dichloromethane/ethyl acetate over 30 min (57 mg, 66%, purity at 254 nm: 99%). ^1^H NMR (300 MHz, DMSO-d^6^) δ = 9.33 (s, 1H, OH); 8.78 (s, 1H, NH); 7.47 (m, 2H, H-18); 6.81 (m, 2H, H-19); 6.03 (d, J = 3.38 Hz, 1H, H-11); 5.64 (d, J = 3.11 Hz, 1H, H-11); 5.31 (s, 1H, H-14); 5.07 (s, 1H, H-14); 4.48 (dd, J = 3.25, 11.13 Hz, 1H, H-7); 3.97 (t, J = 8.77 Hz, 1H, H-2); 3.69 (s, 3H, H-20); 3.24 (m, 1H, H-1); 2.87 (d, J = 8.80 Hz, 1H, H-3); 2.11–2.38 (m, 5H, H-9, H-10, H-6, H-5); 1.64–1.73 (m, 2H, H-10, H-6); 1.34 (s, 3H, H-15); 1.06 (t, J = 13.37 Hz, 1H, H-5). ^13^C NMR (75 MHz, CDCl_3_) δ = 169.44 (Cq, C-13); 157.54 (Cq, C-16); 154.77 (Cq, C-20); 144.44 (Cq, C-8); 139.85 (Cq, C-12); 132.44 (Cq, C-17); 121.12 (CH, C-18); 119.21 (CH2, C-11); 115.73 (CH2, C-14); 113.60 (CH, C-19); 79.78 (CH, C-2); 63.93 (CH, C-7); 62.76 (CH, C-3); 60.47 (Cq, C-4); 55.16 (CH_3_, C-20); 43.87 (CH, C-1); 36.06 (CH_2_, C-5); 29.45 (CH_2_, C-9); 25.68 (CH_2_, C-6); 24.70 (CH_2_, C-10); 17.72 (CH_3_, C-15). HRMS (TOF, ES+) *m*/*z* [M+H]^+^: calcd. for C_23_H_29_N_2_O_6_ 429.2026; found 429.2025.

3-(4-fluorophenyl)-1-hydroxy-1-[(1S,2S,4R,7R,11S)-4-methyl-8,12-dimethylene-13-oxo-3,14-dioxatricyclo[9.3.0.02,4]tetradecan-7-yl]urea (**8**) was obtained as a white powder after purification by flash chromatography in column 4g, with a gradient from 100% dichloromethane to 70/30 dichloromethane/ethyl acetate over 30 min (46 mg, 55%, purity at 254 nm: 99%). ^1^H NMR (300 MHz, DMSO-d^6^) δ = 9.41 (s, 1H, OH); 9.01 (s, 1H, NH); 7.61 (m, 2H, H-18); 7.06 (m, 2H, H-19); 6.03 (d, J = 3.53 Hz, 1H, H-11′); 5.64 (d, J = 3.23 Hz, 1H, H-11); 5.32 (s, 1H, H-14′); 5.08 (s, 1H, H-14); 4.70 (dd, J = 2.60, 10.93 Hz, 1H, H-7); 3.97 (t, J = 9.22 Hz, 1H, H-2); 3.25 (m, 1H, H-1); 2.93 (d, J = 8.80 Hz, 1H, H-3); 2.14–2.38 (m, 5H, H-5′, H-6′, H-10′, 2xH-9); 1.64–1.76 (m, 2H, H-10, H-6); 1.35 (s, 3H, H-15); 1.06 (t, J = 13.27 Hz, 1H, H-5). ^13^C NMR (75 MHz, DMSO-d^6^) δ = 169.39 (Cq, C-13); 159.20 (d, J = 238.68 Hz, Cq, C-20); 157.27 (Cq, C-16); 144.35 (Cq, C-8); 139.83 (Cq, C-12); 135.82 (d, J = 2.42 Hz, Cq, C-17); 121.16 (d, J = 7.63 Hz, CH, C-18); 119.16 (CH_2_, C-11); 115.78 (CH_2_, C-14); 114.99 (d, J = 21.64 Hz, CH, C-19); 79.73 (CH, C-2); 63.81 (CH, C-7); 62.73 (CH, C-3); 60.42 (Cq, C-4); 43.81 (CH, C-1); 36.99 (CH_2_, C-5); 29.35 (CH_2_, C-9); 25.66 (CH_2_, C-6); 24.63 (CH_2_, C-10); 17.70 (CH_3_, C-15). HRMS (TOF, ES+) *m*/*z* [M+]^+^: calcd. for C_22_H_26_FN_2_O_5_ 417.1826; found 417.1810.

3-(1,3-benzodioxol-5-yl)-1-hydroxy-1-[(1S,2S,4R,7R,11S)-4-methyl-8,12-dimethylene-13-oxo-3,14-dioxatricyclo[9.3.0.02,4] tetradecan-7-yl]urea (**10**) was obtained as a white powder after purification by flash chromatography in column 4g, with a gradient from 100% dichloromethane to 70/30 dichloromethane/ethyl acetate over 30 min (30 mg, 34%, purity at 254 nm: 95%). ^1^H NMR (300 MHz, DMSO-d^6^) δ = 9.36 (s, 1H, OH); 8.83 (s, 1H, NH); 7.27 (d, J = 2.05 Hz, 1H, H-18); 7.02 (dd, J = 2.03, 8.29 Hz, 1H, H-23); 6.77 (d, J = 8.39 Hz, 1H, H-22); 6.03 (d, J = 3.41 Hz, 1H, H-11′); 5.93 (s, 2H, H-20); 5.64 (d, J = 3.05 Hz, 1H, H-11); 5.31 (s, 1H, H-14′); 5.07 (s, 1H, H-14); 4.67 (dd, J = 3.00, 11.57 Hz, 1H, H-7); 3.97 (t, J = 9.06 Hz, 1H, H-2); 3.23 (m, 1H, H-1); 2.93 (d, J = 8.72 Hz, 1H, H-3); 2.11–2.35 (m, 5H, H-5′, H-6′, H-10′, 2xH-9); 1.65–1.72 (m, 2H, H-10, H-6); 1.35 (s, 3H, H-15); 1.06 (t, J = 12.08 Hz, 1H, H-5). ^13^C NMR (75 MHz, DMSO-d^6^) δ = 169.35 (Cq, C-13); 157.28 (Cq, C-16); 146.85 (Cq, C-19); 144.32 (Cq, C-8); 142.25 (Cq, C-21); 139.80 (Cq, C-12); 133.77, (Cq, C-17); 119.12 (CH_2_, C-11); 115.70 (CH_2_, C-14); 112.14 (CH, C-23); 107.70 (CH, C-22); 101.72 (CH, C-18); 100.71 (CH_2_, C-20); 79.72 (CH, C-2); 63.83 (CH, C-7); 62.70 (CH, C-3); 60.39 (Cq, C-4); 43.85 (CH, C-1); 36.01 (CH_2_, C-5); 29.41 (CH_2_, C-9); 25.63 (CH_2_, C-6); 24.66 (CH_2_, C-10); 17.67 (CH_3_, C-15). HRMS (TOF, ES+) *m*/*z* [M+H]^+^: calcd. for C_23_H_27_N_2_O_7_ 443.1818; found 443.1812.

3-(3-ethoxypropyl)-1-hydroxy-1-[(1S,2S,4R,7R,11S)-4-methyl-8,12-dimethylene-13-oxo-3,14-dioxatricyclo[9.3.0.02,4]tetradecan-7-yl]urea (**11**) was obtained as a white powder after purification by flash chromatography in column 4g, with a gradient from 100% dichloromethane to 40/50/10 dichloromethane/ethyl acetate/methanol over 30 min (20 mg, 24%, purity at 254 nm: 100%). ^1^H NMR (300 MHz, DMSO-d^6^) δ = 8.90 (s, 1H, OH); 6.89 (t, J = 5.85 Hz, 1H, NH); 6.02 (d, J = 3.30 Hz, 1H, H-11′); 5.64 (d, J = 3.15 Hz, 1H, H-11); 5.24 (s, 1H, H-14′); 5.03 (s, 1H, H-14); 4.54 (dd, J = 2.36, 10.60 Hz, 1H, H-7); 3.95 (t, J = 9.20 Hz, 1H, H-2); 3.34–3.41 (m, 4H, H-20,H-19); 3.24 (m, 1H, H-1); 3.08 (q, J = 6.71 Hz, 2H, H-17); 2.89 (d, J = 8.57 Hz, 1H, H-3); 2.08–2.37 (m, 5H, H-5′, H-6′, H-10′, 2xH-9); 1.56–1.65 (m, 4H, H-10, H-6, 2xH-18); 1.32 (s, 3H, H-15); 1.07 (m, 4H, H-5, H-21). ^13^C NMR (75 MHz, DMSO-d^6^) δ = 169.35 (Cq, C-13); 160.32 (Cq, C-16); 144.54 (Cq, C-8); 139.80 (Cq, C-12); 119.12 (CH_2_, C-11); 115.70 (CH_2_, C-14); 79.72 (CH, C-2); 67.98 (CH_2_, C-19); 65.29 (CH_2_, C-20); 64.37 (CH, C-7); 62.70 (CH, C-3); 60.40 (Cq, C-4); 43.86 (CH, C-1); 37.20 (CH_2_, C-17) 36.08 (CH_2_, C-5); 29.94 (2xCH_2_, C-9, C-18); 25.43 (CH_2_, C-6); 24.75 (CH_2_, C-10); 17.65 (CH_3_, C-15); 15.10 (CH_3_, C-21). HRMS (TOF, ES+) *m*/*z* [M+H]^+^: calcd. for C_21_H_33_N_2_O_6_ 409.2339; found 409.2329. 

*tert*-butyl-N-hydroxy-N-[(1S,2S,4R,7R,11S)-4-methyl-8,12-dimethylene-13-oxo-3,14-dioxatricyclo[9.3.0.02,4]tetradecan-7-yl]carbamate (**12**) was obtained as a white powder after purification by flash chromatography in column 4g, with a gradient from 100% dichloromethane to 70/30 dichloromethane/ethyl acetate over 30 min (42 mg, 55%, purity at 254 nm: 96%). ^1^H NMR (300 MHz, DMSO-d^6^) δ = 8.95 (s, 1H, OH); 6.03 (d, J = 3.36 Hz, 1H, H-11); 5.65 (d, J = 3.07 Hz, 1H, H-11′); 5.27 (s, 1H, H-14); 5.04 (s, 1H, H-14′); 4.46 (dd, J = 3.07, 10.98 Hz, 1H, H-7); 3.96 (t, J = 9.19 Hz, 1H, H-2); 3.23 (m, 1H, H-3); 2.88 (d, J = 8.77 Hz, 1H, H-1); 2.26–2.39 (m, 2H, H-9, H-10); 2.10–2.20 (m, 3H, H-9′, H-6, H-5′); 1.61–1.68 (m, 2H, H-6′, H-10′); 1.40 (s, 9H, H-Boc); 1.33 (s, 3H, H-15) 1.02 (t, J = 12.541 Hz, 1H, H-5). ^13^C NMR (75 MHz, DMSO-d^6^) δ = 169.34 (Cq, C-13); 155.40 (Cq, C-16); 144.50 (Cq, C-8); 139.77 (Cq, C-12); 119.18 (CH2-11); 115.55 (CH2-14); 79.66 (CH, C-2); 64.56 (CH, C-7); 62.88 (CH, C-1); 60.35 (Cq, C-4); 43.51 (CH, C-3); 35.69 (CH_2_, C-5); 28.67 (CH_2_, C-9); 28.08 (CH_3_-Boc); 25.65 (CH_2_, C-6); 24.53 (CH_2_, C-10); 17.70 (CH_3_-15). Quaternary carbon C-18 is not shown. HRMS (TOF, ES+) *m*/*z* [M+NH_4_]^+^: calcd. for C_20_H_33_N_2_O_6_ 397.2339; found 397.2343.

Allyl-N-hydroxy-N-[(1S,2S,4R,7R,11S)-4-methyl-8,12-dimethylene-13-oxo-3,14-dioxatricyclo [9.3.0.02,4]tetradecan-7-yl]carbamate (**13**) was obtained as a white powder after purification by flash chromatography in column 4g, with a gradient from 100% dichloromethane to 70/30 dichloromethane/ethyl acetate over 30 min (37 mg, 50%, purity at 254 nm: 100%). ^1^H NMR (300 MHz, DMSO-d^6^) δ = 9.25 (s, 1H, OH); 6.03 (d, 1H, J = 3.41 Hz, H-11); 5.92 (m, 1H, H-18); 5.62 (d, 1H, J = 3.12 Hz, H-11′); 5.26–5.33 (m, 3H, H-14, 2xH-19); 5.06 (s, 1H, H-14′); 4.52–4.55 (m, 3H, 2xH-17, H-7); 3.96 (t, J = 9.23 Hz, 1H, H-2); 3.24 (m, 1H, H-1); 2.90 (d, J = 8.98 Hz, 1H, H-3); 2.29–2.36 (m, 2H, H-9, H-10); 2.11–2.21 (m, 3H, H-9′, H-6, H-5′); 1.64–1.77 (m, 2H, H-6′, H-10′); 1.35 (s, 3H, H-15); 1.05 (t, J = 13.08 Hz, 1H H-5). ^13^C NMR (75 MHz, DMSO-d^6^) δ = 169.37 (Cq, C-13); 155.80 (Cq, C-16); 144.23 (Cq, C-8); 139.79 (Cq, C-12); 133.23 (CH, C-18); 119.15 (CH_2_, C-11); 117.45 (CH_2_, C-19); 115.98 (CH_2_, C-14); 79.64 (CH, C-2); 65.48 (CH, C-17); 64.67 (Cq, C-7); 62.82 (CH, C-3); 60.34 (Cq, C-4); 43.49 (CH, C-1); 35.58 (CH_2_, C-5); 28.60 (CH_2_, C-9); 25.59 (CH_2_, C-6); 24.35 (CH_2_, C-10); 17.71 (CH_3_, C-15). HRMS (TOF, ES+) *m*/*z* [M+H]^+^: calcd. for C1_9_H_26_NO_6_ 364.1760; found 364.1773.

Benzyl-N-hydroxy-N-[(1S,2S,4R,7R,11S)-4-methyl-8,12-dimethylene-13-oxo-3,14-dioxatricyclo[9.3.0.02,4]tetradecan-7-yl]carbamate (**14**) was obtained as a white powder after purification by flash chromatography in column 4g, with a gradient from 100% dichloromethane to 70/30 dichloromethane/ethyl acetate over 30 min (64 mg, 77%, purity at 254 nm: 98%). ^1^H NMR (300 MHz, DMSO-d^6^) δ = 9.27 (s, 1H, OH); 7.31–7.38 (m, 5H-Ar); 6.03 (d, J = 3.43 Hz, 1H, H-11); 5.64 (d, J = 3.05 Hz, 1H, H-11); 5.29 (s, 1H, H-14); 5.09 (d, J = 3.15 Hz, 2H, H-17); 5.06 (s, 1H, H-14); 4.56 (dd, J = 3.05, 10.88 Hz, H-7); 3.96 (t, J = 9.24 Hz, 1H, H-2); 3.22 (m, 1H, H-1); 2.89 (d, J = 8.87 Hz, 1H, H-3); 2.29–2.36 (m, 2H, H-9, H-10); 2.11–2.20 (m, 3H, H-6, H-5, H-9); 1.63–1.71 (m, 2H, H-6, H-10); 1.34 (s, 3H, H-15); 1.05 (t, J = 12.57 Hz, 1H, H-5). ^13^C NMR (75 MHz, DMSO-d^6^) δ = 169.36 (Cq, C-13); 155.95 (Cq, C-16); 144.21 (Cq, C-8); 139.78 (Cq, C-12); 136.62 (Cq, C-18); 128.39 (CH, 2 C-Ar); 127.98 (CH, C-para Ar); 127.89 (CH, 2 C-Ar); 119.17 (CH_2_, C-11); 116.03 (CH_2_, C-14); 79.64 (CH, C-2); 66.46 (CH_2_, C-17); 64.66 (CH, C-7); 62.82 (CH, C-3); 60.33 (Cq, C-4); 43.44 (CH, C-1); 35.57 (CH_2_, C-5); 28.59 (CH_2_, C-9); 25.59 (CH_2_, C-6); 24.43 (CH_2_, C-10); 17.70 (CH_3_, C-15). HRMS (TOF, ES+) *m*/*z* [M+NH_4_]^+^: calcd. for C_23_H_31_N_2_O_6_ 431.2182; found 431.2187.

2,2,2-trichloroethyl-N-hydroxy-N-[(1S,2S,4R,7R,11S)-4-methyl-8,12-dimethylene-13-oxo-3,14-dioxatricyclo[9.3.0.02,4]tetradecan-7-yl]carbamate (**15**) was obtained as a white powder after purification by flash chromatography in column 4g, with a gradient from 100% dichloromethane to 80/20 dichloromethane/ethyl acetate over 30 min (55 mg, 60%, purity at 215 nm: 100%). ^1^H NMR (300 MHz, DMSO-d^6^) δ = 9.60 (s, 1H, OH); 6.03 (d, J = 3.44 Hz, 1H, H-11); 5.66 (d, J = 3.17 Hz, 1H, H-11); 5.35 (s, 1H, H-14); 5.12 (s, 1H, H-14); 4.88 (m, 2H, H-17); 4.59 (dd, J = 3.07 Hz, 1H, H-7); 3.98 (t, J = 9.10 Hz, 1H, H-2); 3.22 (m, 1H, H-1); 2.90 (d, J = 8.91 Hz, 1H, H-3); 2.13–2.35 (m, 5H, H-5, H-6, H-9, H10); 1.65–1.76 (m, 2H, H-6, H-10); 1.34 (s, 3H, H-15); 1.05 (t, J = 12.63 Hz, 1H, H-5). ^13^C NMR (75 MHz, DMSO-d^6^) δ = 169.80 (Cq, C-13); 154.35 (Cq, C-16); 144.13 (Cq, C-12); 140.19 (Cq, C-8); 119.72 (CH_2_, C-11); 116.96 (CH_2_, C-14); 96.25 (Cq, C-16); 80.10 (CH, C-2); 74.60 (CH_2_, C-17); 65.52 (CH, C-7); 63.31 (CH, C-3); 60.75 (Cq, C-4); 44.08 (CH, C-1); 36.07 (CH_2_, C-5); 29.15 (CH_2_, C-9); 26.02 (CH_2_, C-6); 25.15 (CH_2_, C-10); 18.12 (CH_3_, C-15). HRMS (TOF, ES+) *m*/*z* [M+H]+: calcd. for C_18_H_23_C_l3_NO_6_ 454.0584; found 454.0591.

(4-fluorophenyl)-N-hydroxy-N-[(1S,2S,4R,7R,11S)-4-methyl-8,12-dimethylene-13-oxo-3,14-dioxatricyclo[9.3.0.02,4]tetradecan-7-yl]carbamate (**16**) was obtained as a slightly yellow powder after purification by flash chromatography in column 4g, with a gradient from 100% dichloromethane to 70/30 dichloromethane/ethyl acetate over 30 min (28 mg, 33%, purity at 254 nm: 98%). ^1^H NMR (300 MHz, DMSO-d^6^) δ = 9.67 (s, 1H); 7.11–7.25 (m, 4H, H-Ar); 6.04 (d, J = 3.50 Hz, 1H, H-11); 5.66 (d, J = 3.22 Hz, 1H, H-11′); 5.38 (s, 1H, H-14); 5.15 (s, 1H, H-14′); 4.63 (dd, J = 3.01, 11.46 Hz, 1H, H-7); 3.98 (t, J= 9.06 Hz, 1H, H-2); 3.25 (m, 1H, H-1); 2.93 (d, J = 8.93 Hz, 1H, H-3); 2.15–2.42 (m, 5H, H-5, H-6, H-10, H-9); 1.67–1.84 (m, 2H, H-6′, H-10′); 1.36 (s, 3H, H-15); 1.10 (t, J = 12.62 Hz, 1H, H-5′). ^13^C NMR (75 MHz, DMSO-d^6^) δ = 169.38 (Cq, C-13); 159.29 (d, J = 241.78 Hz, Cq, C-20); 147.06 (d, J = 2.99 Hz, Cq, C-17); 143.90 (Cq, C-8); 139.77 (Cq, C-12); 123.51 (d, J = 8.78 Hz, 2xCH, C-18); 119.17 (CH_2_, C-11); 116.39 (CH_2_, C-14); 115.95 (d, J = 23.10 Hz, 2xCH, C-19); 79.61 (CH, C-2); 65.04 (CH, C-7); 62.70 (CH, C-3); 60.34 (Cq, C-4); 43.64 (CH, C-1); 35.53 (CH_2_, C-5); 28.63 (CH_2_, C-9); 25.60 (CH_2_, C-6); 24.29 (CH_2_, C-10); 17.69 (CH_3_, C-15). HRMS (TOF, ES+) *m*/*z* [M+H]^+^: calcd. for C_22_H_25_FNO_6_ 418.1654; found 418.1666.

Phenyl-N-hydroxy-N-[(1S,2S,4R,7R,11S)-4-methyl-8,12-dimethylene-13-oxo-3,14-dioxatricyclo[9.3.0.02,4]tetradecan-7-yl]carbamate (**17**) was obtained as a white powder after purification by flash chromatography in column 4g, with a gradient from 100% dichloromethane to 80/20 dichloromethane/ethyl acetate over 30 min (37 mg, 45%, purity at 254 nm: 99%). ^1^H NMR (300 MHz, DMSO-d^6^) δ = 9.64 (s, 1H, OH); 7.40 (m, 2H, H-18); 7.23 (m, 1H, H-20); 7.09 (m, 2H, H-19); 6.03 (d, J = 3.44 Hz, 1H, H-11′); 5.66 (d, J = 3.13 Hz, 1H, H-11); 5.38 (s, 1H, H-14′); 5.15 (s, 1H, H-14); 4.64 (dd, J = 3.06, 11.38 Hz, 1H, H-7); 3.98 (t, J = 9.08 Hz, 1H, H-2); 3.26 (m, 1H, H-1); 2.93 (d, J = 9.08 Hz, 1H, H-3); 2.15–2.43 (m, 5H, H-5′, H-6′, H-10′, H-9); 1.65–1.84 (m, 2H, H-10, H-6); 1.36 (s, 3H, H-15); 1.11 (t, J = 12.49 Hz, 1H, H-5). ^13^C NMR (75 MHz, DMSO-d^6^) δ = 169.37 (Cq, C-13); 154.06 (Cq, C-16); 151.00 (Cq, C-17) 143.97 (Cq, C-8); 139.78 (Cq, C-12); 129.40 (2xCH, C-18); 125.36 (CH, C-20); 121.70 (2xCH, C-19); 119.16 (CH_2_, C-11); 116.35 (CH_2_, C-14); 79.61 (CH, C-2); 65.01 (CH, C-7); 62.71 (CH, C-3); 60.34 (Cq, C-4); 43.60 (CH, C-1); 35.53 (CH_2_, C-5); 28.61 (CH_2_, C-9); 25.62 (CH_2_, C-6); 24.27 (CH_2_, C-10); 17.69 (CH_3_, C-15). HRMS (TOF, ES+) *m*/*z* [M+H]^+^: calcd. for C_22_H_26_NO 400.1760; found 400.1747.

Isobutyl-N-hydroxy-N-[(1S,2S,4R,7R,11S)-4-methyl-8,12-dimethylene-13-oxo-3,14-dioxatricyclo[9.3.0.02,4]tetradecan-7-yl]carbamate (**18**) was obtained as a white powder after purification by flash chromatography in column 4g, with a gradient from 100% dichloromethane to 70/30 dichloromethane/ethyl acetate over 30 min (50 mg, 65%, purity at 215 nm: 98%). ^1^H NMR (300 MHz, DMSO-d^6^) δ = 9.16 (s, 1H, OH); 6.03 (d, J = 3.39 Hz, 1H, H-11); 5.65 (d, J = 3.13 Hz, 1H, H-11); 5.28 (s, 1H, H-14); 5.06 (s, 1H, H-14); 4.51 (dd, J = 2.90, 10.90 Hz, H-7); 3.97 (t, J = 9.16 Hz, 1H, H-2); 3.79 (d, J = 6.03 Hz, 2H, H-17); 3.23 (m, 1H, H-1); 2.89 (d, J = 8.93 Hz, 1H, H-3); 2.29–2.38 (m, 2H, H-9, H-10); 2.11–2.23 (m, 3H, H-6, H-5, H-9); 1.85 (m, 1H, H-18); 1.64–1.71 (m, 2H, H-6, H-10); 1.34 (s, 3H, H-15); 1.04 (t, J = 12.32 Hz, 1H, H-5); 0.87 (d, J = 6.67 Hz, 6H, H-19, H-20). ^13^C NMR (75 MHz, DMSO-d^6^) δ = 169.83 (Cq, C-13); 156.72 (Cq, C-16); 144.75 (Cq, C-8); 140.22 (Cq, C-12); 119.64 (CH_2_, C-11); 116.34 (CH_2_, C-14); 80.10 (CH, C-2); 71.40 (CH_2_, C-17) 65.18 (CH, C-7); 63.32 (CH, C-3); 60.80 (Cq, C-4); 43.93 (CH, C-1); 36.06 (CH_2_, C-5); 29.05 (CH_2_, C-9); 28.03 (CH, C-18); 26.06 (CH_2_, C-6); 24.88 (CH_2_, C-10); 19.32 (CH_3_, C-19, C-20); 18.15 (CH_3_, C-15). HRMS (TOF, ES+) *m*/*z* [M+H]^+^: calcd. for C_20_H_30_NO_6_ 380.2073; found 380.2045.

Ethyl-N-hydroxy-N-[(1S,2S,4R,7R,11S)-4-methyl-8,12-dimethylene-13-oxo-3,14-dioxatricyclo[9.3.0.02,4]tetradecan-7-yl]carbamate (**19**) was obtained as a white powder after purification by flash chromatography in column 4g, with a gradient from 100% dichloromethane to 60/40 dichloromethane/ethyl acetate over 30 min (30 mg, 42%, purity at 254 nm: 98%). ^1^H NMR (300 MHz, DMSO-d^6^) δ = 9.17 (s, 1H, OH); 6.02 (d, J = 3.40 Hz, 1H, H-11); 5.64 (d, J = 3.14 Hz, 1H, H-11′); 5.28 (s, 1H, H-14); 5.05 (s, 1H, H-14′); 4.51 (dd, J = 2.99, 10.85 Hz, 1H, H-7); 3.93–4.08 (m, 3H, H-2, 2xH-17); 3.23 (m, 1H, H-1); 2.89 (d, J = 8.89 Hz, H-3); 2.28–2.38 (m, 2H, H-9, H-10); 2.08–2.21 (m, 3H, H-6, H-5, H-9′); 1.63–1.70 (m, 2H, H-6′, H-10′); 1.33 (s, 3H, H-15); 1.17 (t, J = 7.09 Hz, 3H, H-18); 1.04 (t, J = 12.66 Hz, 1H, H-5′). ^13^C NMR (75 MHz, DMSO-d^6^) δ = 169.47 (Cq, C-13); 156.30 (Cq, C-16); 144.35 (Cq, C-8); 139.83 (Cq, C-12); 119.25 (CH_2_, C-11); 115.95 (CH_2_, C-14); 79.73 (CH, C-2); 64.64 (CH, C-7); 62.90 (CH, C-3); 61.10 (CH_2_-C-17); 60.43 (Cq, C-4); 43.53 (CH, C-1); 35.65 (CH_2_, C-5); 28.66 (CH_2_, C-9); 25.64 (CH_2_, C-6); 24.42 (CH_2_, C-10); 17.76 (CH_3_, C-15); 14.65 (CH3, C-18). HRMS (TOF, ES+) *m*/*z* [M+H]^+^: calcd. for C_18_H_26_NO_6_ 352.1760; found 352.1764.

1,3-benzodioxol-5-yl-N-hydroxy-N-[(1S,2S,4R,7R,11S)-4-methyl-8,12-dimethylene-13-oxo-3,14-dioxatricyclo[9.3.0.02,4]tetradecan-7-yl]carbamate (**20**) was obtained as a white powder after purification by flash chromatography in column 4g, with a gradient from 100% dichloromethane to 80/20 dichloromethane/ethyl acetate over 30 min (37 mg, 41%, purity at 254 nm: 96%). ^1^H NMR (300 MHz, DMSO-d^6^) δ = 9.61 (s, 1H, OH); 6.88 (d, 1H, J = 8.46 Hz, H-22); 6.73 (d, 1H, J = 2.37 Hz, H-18); 6.53 (dd, 1H, J = 2.37, 8.46 Hz, H-21); 6.02–6.05 (m, 3 H, H-20, H-11); 5.66 (d, 1H, J = 3.47 Hz, H-11′); 5.37 (s, 1H, H-14); 5.14 (s, 1H, 14′); 4.60 (dd, 1H, J = 10.94, 2.43 Hz, H-7); 3.97 (t, 1H, J = 8.99 Hz, H-2); 3.25 (m, 1H, H-1); 2.90 (d, 1H, J = 8.76 Hz, H-3); 2.13–2.43 (m, 5H, H-9, H-9′, H-10, H-6, H-5); 1.65–1.85 (m, 2H, H-10′, H-6′); 1.36 (s, 3H, H-15); 1.10 (m, 1H, H-5′). ^13^C NMR (75 MHz, DMSO-d^6^) δ = 169.38 (Cq, C-13); 154.27 (Cq, C-16); 147.49 (Cq, C-19); 145.29 (Cq, C-17); 144.61 (Cq, C-21);143.93 (Cq, C-8); 139.78 (Cq, C-12); 119.17 (CH_2_, C-11); 116.34 (CH_2_, C-14); 114.09 (CH, C-23); 107.82 (CH, C-22); 103.94 (CH, C-18); 101.66 (CH_2_, C-20); 79.02 (CH, C-2); 65.01 (CH, C-7); 62.70 (CH, C-3); 60.35 (Cq, C-4); 43.62 (CH, C-1); 35.53 (CH_2_, C-5); 28.63 (CH_2_, C-9); 25.60 (CH_2_, C-6); 24.28 (CH_2_, C-10); 17.69 (CH_3_, C-15). HRMS (TOF, ES+) *m*/*z* [M+H]^+^: calcd. for C_23_H_26_NO_8_ 444.1658; found 444.1629.

2-methoxyethyl-N-hydroxy-N-[(1S,2S,4R,7R,11S)-4-methyl-8,12-dimethylene-13-oxo-3,14-dioxatricyclo[9.3.0.02,4]tetradecan-7-yl]carbamate (**21**) was obtained as a white powder after purification by flash chromatography in column 4g, with a gradient from 100% dichloromethane to 70/30 dichloromethane/ethyl acetate over 30 min (30 mg, 40%, purity at 254 nm: 99%). ^1^H NMR (300 MHz, DMSO-d^6^) δ = 9.23 (s, 1H, OH); 6.03 (d, J = 3.45 Hz, 1H, H-11); 5.65 (d, J = 3.06 Hz, 1H, H-11′); 5.29 (s, 1H, H-14); 5.06 (s, 1H, H-14′); 4.51 (dd, J = 2.87, 10.52 Hz, 1H, H-7); 4.11 (m, 2H, H-17) 3.97 (t, J = 9.18 Hz, H-2); 3.50 (t, J = 4.78 Hz, 2H, H-18); 3.22–3.26 (m, 4H, H-1, H-19); 2.89 (d, J = 8.99 Hz, H-3); 2.28–2.38 (m, 2H, H-9, H-10); 2.08–2.21 (m, 3H, H-6, H-5, H-9′); 1.63–1.73 (m, 2H, H-6′, H-10′); 1.34 (s, 3H, H-15); 1.04 (t, J = 12.63 Hz, 1H, H-5′). ^13^C NMR (75 MHz, DMSO-d^6^) δ = 169.83 (Cq, C-13); 156.55 (Cq, C-16); 144.76 (Cq, C-8); 140.24 (Cq, C-12); 119.66 (CH2, C-11); 116.36 (CH2, C-14); 80.16 (CH, C-2); 70.55 (CH_2_, C-18); 64.96 (CH, C-7); 64.65 (CH_2_, C-17); 63.34 (CH, C-3); 60.79 (Cq, C-4); 58.46 (CH_3_, C-19); 43.87 (CH, C-1); 36.06 (CH_2_, C-5); 29.06 (CH_2_, C-9); 26.01 (CH_2_, C-6); 25.06 (CH_2_, C-10); 18.17 (CH_3_, C-15). HRMS (TOF, ES+) *m*/*z* [M+H]^+^: calcd. for C_19_H_28_NO_7_ 382.1866; found 382.1875.

Tetrahydropyran-4-yl-N-hydroxy-N-[(1S,2S,4R,7R,11S)-4-methyl-8,12-dimethylene-13-oxo-3,14-dioxatricyclo[9.3.0.02,4]tetradecan-7-yl]carbamate (**22**) was obtained as a white powder after purification by flash chromatography in column 4g, with a gradient from 100% dichloromethane to 70/30 dichloromethane/ethyl acetate over 30 min (52 mg, 63%, purity at 254 nm: 95%). ^1^H NMR (300 MHz, DMSO-d^6^) δ = 9.18 (s, 1H, OH); 6.03 (d, J = 3.47 Hz, 1H, H-11); 5.65 (d, J = 3.16 Hz, 1H, H-11′); 5.28 (s, 1H, H-14); 5.05 (s, 1H, H-14′); 4.73 (m, 1H, H-17); 4.52 (dd, J = 3.00, 10.75 Hz, 1H, H-7); 3.97 (t, J = 9.17 Hz, 1H, H-2); 3.79 (m, 2H, H-19); 3.42 (m, 2H, H-20 under water peak); 3.23 (m, 1H, H-1); 2.89 (d, J = 9.01 Hz, H-3); 2.11–2.38 (m, 5H, H-6, H-5, 2xH-9, H-10); 1.84 (m, 2H, H-18); 1.63–1.70 (m, 2H, H-6′, H-10′); 1.52 (m, 2H, H-21); 1.33 (s, 3H, H-15); 1.04 (t, J = 14.57 Hz, 1H, H-5′). ^13^C NMR (75 MHz, DMSO-d^6^) δ = 169.83 (Cq, C-13); 155.92 (Cq, C-16); 144.73 (Cq, C-8); 140.23 (Cq, C-12); 119.65 (CH2, C-11); 116.33 (CH2, C-14); 80.10 (CH, C-2); 70.62 (CH, C-17); 65.08 (CH, C-7); 64.99 (2xCH_2_, C-19, C-20); 63.32, (CH, C-3); 60.81 (Cq, C-4); 43.96 (CH, C-1); 36.05 (CH_2_, C-5); 32.36 (2xCH_2_, C-18, C-21); 29.08 (CH_2_, C-9); 26.08 (CH_2_, C-6); 24.90 (CH_2_, C-10); 18.16 (CH_3_, C-15). HRMS (TOF, ES+) *m*/*z* [M+NH_4_]^+^: calcd. for C_21_H_33_N_2_O_7_ 425.2288; found 425.2279.

#### 3.1.4. Synthesis of 1-hydroxy-1-[(1S,2S,4R,7R,11S)-4-methyl-8,12-dimethylene-13-oxo-3,14-dioxatricyclo[9.3.0.02,4]tetradecan-7-yl]urea (**9**)

In a double-neck flask, CuCl (0.05 eq., 1 mg) and pyridine (0.125 eq., 2 µL) were added to the parthenolide (50 mg) and hydroxylamine derivative (1.1 eq.). Then, 2.2 mL of THF was added. The reaction mixture was stirred at room temperature while being exposed to the air. The reaction was monitored using TLC. After 24 h or 48 h, the reaction was stopped with a solution of EDTA (0.5 M, pH 7.0), diluted with ethyl acetate, and stirred until the color no longer persisted in the organic layer. Then, the water phase was extracted three times with ethyl acetate. The organic phase was dried over MgSO_4_ and concentrated under reduced pressure to give a crude residue. Purification by reversed-phase chromatography in column 15g, with a gradient from water/acetonitrile (90/10) to acetonitrile (100%) over 30 min afforded the desired compound (9) as a white powder (59 mg, 91%, purity at 254 nm: 99%). ^1^H NMR (300 MHz, DMSO-d^6^) δ = 8.97 (s, 1H, OH); 6.28 (s, 2H, NH2); 6.03 (d, J = 3.42 Hz, 1H, H-11); 5.64 (d, J = 3.11 Hz, 1H); 5.26 (s, 1H, H-14); 5.03 (s, 1H, H-14); 4.57 (dd, J = 2.63, 11.23 Hz, 1H, H-7); 3.95 (t, J = 9.41 Hz, 1H, H-2); 3.23 (m, 1H, H-1); 2.90 (d, J = 8.89 Hz, 1H, H-3); 2.23–2.33 (m, 3H, 2xH-9, H-10); 2.04–2.20 (m, 2H, H-5, H-6) 1.57–1.71 (m, 2H, H-6, H-10); 1.32 (s, 3H, H-15); 1.00 (t, J = 12.97 Hz, 1H, H-5). ^13^C NMR (75 MHz, DMSO-d^6^) δ = 169.41 (Cq, C-13); 161.18 (Cq, C-16); 144.65 (Cq, C-8); 139.86 (Cq, C-12); 119.16 (CH_2_, C-11); 115.30 (CH_2_, C-14); 79.76 (CH, C-2); 63.52 (CH, C-7); 62.72 (CH, C-3); 60.46 (Cq, C-4); 43.93 (CH, C-1); 35.10 (CH_2_, C-5); 28.50 (CH_2_, C-9); 25.51 (CH_2_, C-6); 24.78 (CH_2_, C-10); 17.68 (CH_3_, C-15). HRMS (TOF, ES+) *m*/*z* [M+H]^+^: calcd. for C_16_H_23_N_2_O_5_ 323.1607; found 323.1610.

#### 3.1.5. Synthesis of (1S,2S,4R,7Z,11S)-7-hydroxyimino-4-methyl-8,12-dimethylene-3,14-dioxatricyclo[9.3.0.02,4] tetradecan-13-one (**23**)

In a double-neck flask, CuCl (0.05 eq., 1 mg) and pyridine (0.125 eq., 2 µL) were added to the parthenolide (50 mg) and hydroxylamine derivative (1.1 eq.). Then, 2.2 mL of THF was added. The reaction mixture was stirred at room temperature under O_2_ (in a balloon). The reaction was monitored using TLC. After 24 h or 48 h, the reaction was stopped with a solution of EDTA (0.5 M, pH 7.0), diluted with ethyl acetate, and stirred until the color no longer persisted in the organic layer. Then, the water phase was extracted three times with ethyl acetate. The organic phase was dried over MgSO_4_ and concentrated under reduced pressure to give a crude residue. Purification by reversed-phase chromatography in column 4g, with a gradient from water/acetonitrile (90/10) to acetonitrile (100%) over 30 min afforded the desired compound (23) as a white powder (10 mg, 18%, purity at 254 nm: 99%). ^1^H NMR (300 MHz, DMSO-d^6^) δ = 11.30 (s, OH); 6.09 (d, J = 3.67 Hz, 1H, H-11); 5.76 (d, J = 3.36 Hz, 1H, H-11′) 5.48 (s, 1H, H-14′); 5.39 (s, 1H, H-14); 3.92 (t, J = 9.34 Hz, 1H, H-2); 2.87–3.02 (m, 2H, H-1, H-10); 2.55–2.70 (m, 3H, H-3, H-9, H-10′); 2.45 (m, 1H, H-9′); 2.05–2.18 (m, 2H, H-5, H-6); 1.47–1.61 (m, 2H, H-6′, H-5′); 1.26 (s, 3H, H-15). ^13^C NMR (75 MHz, DMSO-d^6^) δ = 169.28 (Cq, C-13); 156.24 (Cq, C-16); 145.00 (Cq, C-8); 139.83 (Cq, C-12); 119.67 (CH2, C-11); 118.11 (CH_2_, C-14); 81.59 (CH, C-2); 64.04 (CH, C-3); 60.06 (Cq, C-4); 44.14 (CH, C-1); 32.95 (CH_2_, C-5); 32.25 (CH_2_, C-9); 28.55 (CH_2_, C-6); 19.56 (CH_2_, C-10); 17.22 (CH_3_, C-15). HRMS (TOF, ES+) *m*/*z* [M+H]^+^: calcd. for C_15_H_20_NO_4_ 278.1392; found 278.1408.

#### 3.1.6. Synthesis of (1S,2S,4R,7E,11S)-4,8,12-trimethyl-3,14-dioxatricyclo[9.3.0.02,4]tetradec-7-en-13-one (**26**)

In an assay tube, parthenolide (100 mg) was diluted in 8 mL of methanol (0.05 M). The reaction was performed in a H-Cube with a Pd/C cartridge at r.t. in the full H_2_ mode. Three runs were necessary to obtain the full conversion of the parthenolide. The reaction led to the formation of 2 diastereomers. Major diastereomers: 70%; minor diastereomers: 30% (101 mg, 61%, purity at 254 nm: 97%). ^1^H NMR (300 MHz, CDCl_3_) δ = 5.17 (dd, J = 2.53 Hz, 1H); 3.81 (t, J = 9.12 Hz, 1H); 2.70 (d, J = 9.12 Hz, 1H); 2.25–2.42 (m, 3H); 2.09–2.19 (m, 2H); 2.04 (m, 1H); 1.79–1.93 (m, 2H); 1.59–1.70 (m, 8H); 1.11–1.29 (m, 8H).

### 3.2. Biological Evaluation

#### 3.2.1. Strains and Growth Conditions

The *Mycobacterium tuberculosis* strain H37Rv expressing GFP, H37Rv::pJKD6, was generated as described previously (DOI:10.1128/AEM.03677-15) and cultured in Middlebrook 7H9 medium supplemented with 10% OADC, 0.2% glycerol, 0.05% Tween^®^ 80, and 20 µg/mL kanamycin (for maintaining selection on pJKD6). *Staphylococcus aureus* (SH1000) was kindly provided by Simon J. Foster (the University of Sheffield) and grown in cation-adjusted Mueller Hinton II broth.

#### 3.2.2. Determination of MICs

The minimal inhibitory concentrations (MICs) of the compounds against *M. tuberculosis* were determined using both GFP and resazurin (REMA) as a readout of the bacterial viability. Briefly, a mid-log phase of the H37Rv::pJKD6 cultures was diluted to an OD600 of 0.001 and distributed in a 96-well plate (100 µL) in the presence of serial dilutions of the compounds of interest. Following 7 days of incubation (at 37 °C), bacterial growth was measured based on GFP fluorescence (Ex. at 485 nm; Em. at 510 nm) using an EnSight fluorescence plate reader (Perkin Elmer). Subsequently, 10 µL of resazurin (0.1% *w*/*v*) was added to the same bacteria and incubated (for 18 h at 37 °C), and the bacterial metabolic conversion of resazurin to resorufin was measured based on fluorescence (Ex. at 560 nm; Em. at 590 nm) using the same plate reader. For the GFP fluorescence readout, the data were background-corrected (without a bacterial control) and expressed as a percentage of the GFP signal compared to that for the untreated bacteria, with MIC GFP defined as the concentration that results in less than 2% of the initial GFP signal. For the resazurin (REMA) fluorescence readout, the data were background-corrected (without a bacterial control) and expressed as a percentage of the REMA signal compared to that for the untreated bacteria, with MIC REMA defined as the concentration that results in less than 10% of the initial REMA signal.

The MICs of the compounds against *S. aureus* were determined using only the resazurin (REMA)-based viability readout. The experiment was performed as for *M. tuberculosis*, with the following changes: the bacteria were grown in the presence of the compounds for only 5 h (not 7 days) and were co-incubated with resazurin for 1 h before the fluorescence reading and analysis.

#### 3.2.3. X-ray Structural Determination

Single-crystal X-ray crystallographic analyses were conducted on compound **14**. The diffraction data were obtained using a combination of phi- and omega-scans on a Bruker Apex DUO diffractometer equipped with a Photon 3 hybrid pixel area detector mounted on a four-circle D8 goniometer. Cu Kα radiation (λ = 1.54178 Å) was obtained using an ImuS Incoatec Cu microfocus sealed tube. The data solution was found using SHELXT [33] and refined using SHELXL [34], as implemented in the Olex 2 crystallographic suite for small molecules [35]. The X-ray crystallographic data for compound **14** have been deposited at the Cambridge Crystallographic Data Centre, under the reference number CCDC-2226330, and can be obtained freely at https://www.ccdc.cam.ac.uk/structures/ (accessed on 11 December 2023).

#### 3.2.4. Solubility Determination

The kinetic solubility was determined using LC–MS/MS analysis in PBS at pH 7.4. Briefly, a 10 mM 100% DMSO solution of the compound was diluted 50-fold either in PBS at pH 7.4 (in triplicate) or in DMSO in polypropylene tubes (*n* = 3 for PBS and *n* = 6 for DMSO). The tubes were gently shaken for 24 h at 21 °C. Then, the three PBS tubes and three of the six DMSO tubes were centrifuged for 5 min at 4000 rpm and filtered over 0.45 μm filters (Millex-LH Millipore). Then, 40 μL of each solution was diluted in 160 μL of MeCN and 40 µL of Milli-Q water and transferred to matrix tubes for the LC–MS/MS analysis. The solubility was determined according to the following formula: Solubility (μM) = (AUC (filtered PBS)/AUC (not filtered DMSO)) × 200. The test was validated if (AUC (not filtered DMSO) − AUC (filtered DMSO))/AUC (not filtered DMSO) ≤ 10%.

#### 3.2.5. Cytotoxicity

The cytotoxicity of the compounds to BALB/3T3 cells was determined using live imaging following both Hoechst 33342 and propidium iodide staining. Briefly, BALB/3T3 cells were seeded in a 384-well plate and 24 h later, the compounds were added (0, 12.5, 25, 50, and 100 µM) to the culture medium, as well as Hoechst 33342 and propidium iodide (PI). Then, 24 h and 48 h after the compounds were added, live imaging was performed using an In Cell Analyzer 6000 (STP GE Healthcare, Glascow, UK). The cytotoxicity was defined based on the ratio of the population of dead or dying cells (propidium iodide staining) to the total cell population (Hoechst staining), as determined using Columbus software 2.9.1 (PerkinElmer Informatics, WA, USA). The compounds were tested in triplicate. Carfilzomib (500, 250, 140, and 60 nM) was used as the positive control in this assay.

## 4. Conclusions

In this study, we described the regioselective and stereoselective acylnitroso-ene modification of parthenolide, which provided 19 original compounds that were screened for their antibacterial activities. The reaction proceeded through the Markovnikov addition of an electron-deficient enophile, nitroso species, to an ene, parthenolide. To the best of our knowledge, the mechanism of this reaction has not been fully elucidated yet. Therefore, according to our crystallographic and NOESY data, we were able to suggest a mechanism to explain the regioselectivity and stereoselectivity. Two series of compounds, ureas and carbamates, bearing aromatic and alkyl groups, were synthesized in moderate to good yields. All the compounds were tested against *M. tuberculosis* and *S. aureus*, and *M. tuberculosis*-specific antibacterial activity was found. It should also be noted that the Michael acceptor is necessary for the activity, but further studies are necessary to fully understand how nitroso-ene derivatives inhibit *M. tuberculosis* growth. Advantageously, nitroso-ene derivatives are more water soluble than parthenolide while maintaining activity against *M. tuberculosis*.

## Data Availability

Data is contained within the article and Appendix A.

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
