# Peer review of "Regioselective and Stereoselective Synthesis of Parthenolide Analogs by Acyl Nitroso-Ene Reaction and Their Biological Evaluation against Mycobacterium tuberculosis"

_ijms, 2023, doi:10.3390/ijms242417395_

Round 1

Reviewer 1 Report

Comments and Suggestions for Authors

The submitted paper is focused on the synthesis and biological evaluation of parthenolide analogues. In general the paper is well written and organized. However, before further peer review process some changes could be done to increase the quality of the manuscript.

1) what about the toxicity of new analogues - please address this issue

2) please provide more info (experimental data) on purity of obtained compounds

Reviewer 2 Report

Comments and Suggestions for Authors

The manuscript of Gioia & all  is dealing with the synthesis of parthenolide  analogues and their biological evaluation against Mycobacterium tuberculosis. The chemistry is simple and efficient, and a reaction mechanism that explain regioselectivity and stereoselectivity of reactions is presented. The activity against Mtb is reasonable. In my opinion, the manuscript could be published after a carefully revision. My major concern is about toxicity of compounds: being highly reactive derivatives I wonder what will be the toxicity in regard with the normal cells? Hence, the toxicity studies have to be done.

Comments on the Quality of English Language

Minor

Reviewer 3 Report

Comments and Suggestions for Authors

In the article with the title “Regioselective and stereoselective synthesis of parthenolide analogues, by acylnitroso ene reaction and their biological evaluation against Mycobacterium tuberculosis.” The authors describe on 21 pages the synthesis of new compounds via acylnitroso ene reaction. They were able to obtain a crystal structure of a target compound, which confirmed the results of the NOESY experiments. The configuration of the new chiral carbon atom led to the prediction of the mechanism of this reaction. The biological evaluation of the synthesized compounds was determined for M. tuberculosis and S. aureus.

Comments

1.       In the manuscript there are two different abbreviations for Mycobacterium tuberculosis and Staphylococcus aureus present (e.g. Sa and Mtb). Please use the abbreviations (M. tuberculosis) and (S. aureus) at their first appearance in the document and please use it afterwards also in table descriptions.

2.       Please write names of the bacteria in italics in the whole document (e.g. lines 163, 198)

3.       In tables 1 and 2 please specify what is meant by ND.

4.       It would be nice for the reader if the crystal structure is in colour.

5.       Please give company name, city and country for all reagents and machinery that is described in this document.

Comments on the Quality of English Language

Minor spellcheck.

Reviewer 4 Report

Comments and Suggestions for Authors

In the paper entitled "Regioselective and stereoselective synthesis of parthenolide analogues, by acylnitroso ene reaction and their biological evaluation against Mycobacterium tuberculosis", the authors present the design, synthesis, caracterization and antibacterial (against M. tuberculosis and S. aureus) effect off some paarthenolide derivatives, obtained through a state of the art nitroso-ene reaction.

The chemistry parte of the research/paper is strong, the paper is well organised and written, however, it needs some minor to major changes before acceptance:

- Title: I was wondering of S. aureus should be included, even if the results obtained are weak;

- authors should make sure that pathogens' and plants' names are written correctly (Italic face);

- reference and some results are written in different font style (Calibri perhaps);

- Schemes/figures are sometimes mentioned within the text using Bold face, sometimes not; authors should use one and the same style;

- authors should be careful with compound numbering (19 or 20?);

- Rifampicin was used as pozitive control for both Mtb and Sa? it is unclear...also, the results obtained for rifampicin are not provided, therefore it is unclear how it was used as pozitive control;

- page 10 - line 267: replace "previous published" with "previously published";

- all sentences should start with capital letters (see lines 311, 371, 386, 401, 416, 431, etc.);

- check lines 312 and 316;

- Discussion is a bit short, but some of the information is presented as results;

- I was wondering if bibliography could be more up to date, considering the originality of the results (around 40% of the references cited are more then 10 years old).

Comments on the Quality of English Language

English language fine.

Round 2

Reviewer 2 Report

Comments and Suggestions for Authors

The manuscript could be published now.

Author Response

We would like to thank the reviewer

Reviewer 4 Report

Comments and Suggestions for Authors

In this revised form of the paper entitled "Regioselective and stereoselective synthesis of parthenolide analogues, by acylnitroso ene reaction and their biological evaluation against Mycobacterium tuberculosis.", the authors have made almost all the changes requested, improving thus the quality of their paper.

However, there are still 3 very minor issues to be solved before acceptance:

- regarding the compounds' numbering, see page 3 line 118 (it sais that 19 new compounds have been developed, while conclusion and the authors' reply say 20);

- do not forget to mark in Italic face "Mycobacterium tuberculosis" in the title;

- regarding the authors' reply "the idea was to integrate the results and the discussion in a unique paragraph. So, the paragraph discussion should be the conslusion"....if the Editor is fine with it, it should be OK.

Author Response

We would like to thank again the reviewer for the valuable comments.

Line 244, we replaced twenty by nineteen.

We changed the title with "Mycobacterium tuberculosis" in italic.

Paragraph 3 remains for us a conclusion while the data are presented and discussed in the preceding paragraphs.
